# Seasonal fire danger forecasts for supporting fire prevention management in an eastern Mediterranean environment: the case study of Attica, Greece

Anna Karali[1,2], Konstantinos V. Varotsos[1], Christos Giannakopoulos[1], Panagiotis P. Nastos[2], Maria Hatzaki[2]

[1]Institute for Environmental Research and Sustainable Development, National Observatory of Athens, Athens, 15236, Greece
[2] Laboratory of Climatology and Atmospheric Environment, Section of Geography and Climatology, Department of Geology and Environment, National and Kapodistrian University of Athens, Athens, 15784, Greece

*Correspondence to*: Anna Karali (akarali@noa.gr)

**Abstract.** Forest fires constitute a major environmental and socioeconomic hazard in the Mediterranean. Weather and climate are among the main factors influencing forest fire potential. As fire danger is expected to increase under changing climate, seasonal forecasting of meteorological conditions conductive to fires is of paramount importance for implementing effective fire prevention policies. The aim of the current study is to provide high resolution (~9km) probabilistic seasonal fire danger forecasts, utilizing the Canadian Fire Weather Index (FWI) for Attica region, one of the most fire prone regions in Greece and the Mediterranean, employing the fifth generation ECMWF seasonal forecasting system (SEAS5). Results indicate that, depending on the lead time of the forecast, both FWI and ISI (Initial Spread Index) present statistically significant high discrimination scores and can be considered reliable in predicting above normal fire danger conditions. When comparing year-by-year the fire danger predictions with the historical fire occurrence recorded by the Hellenic Fire Service database, both seasonal FWI and ISI forecasts are skilful in identifying years with high fire occurrences. Overall, fire danger and its subcomponents can potentially be exploited by regional authorities in fire prevention management regarding preparedness and resources allocation.

## 1    Introduction

The Mediterranean region includes more than 25 million hectares of forests and about 50 million hectares of other wooded lands that make vital contributions to rural development, poverty alleviation and food security, as well as to the agriculture, water, tourism, and energy sectors (FAO and Plan Bleu, 2018). The Mediterranean is considered a high fire risk region where fires cause severe environmental and economic losses and even losses of human lives (MedECC, 2020). Severe forest fires have consistently affected Europe since the beginning of the century, especially regarding the five European Mediterranean countries of Portugal, Spain, Italy, Greece, and France which on average collectively account for approximately 85% of the total burnt area in Europe per year (Costa et al. 2020).

Weather and climate, vegetation conditions and composition, as well as human activities play an essential role in fire regimes (Costa et al., 2020). According to Rogers et al. (2020), climate highly affects fuel properties and short-term weather patterns determine fuel moisture and physical conditions necessary for fire spread. Regarding the Mediterranean, the combination of extreme drought with extreme winds or heatwaves has been identified as crucial factor for the occurrence of wildfires (Ruffault et al., 2020). Under changing climatic conditions, future fire danger as well as the frequency and the extent of large wildfires

are expected to increase throughout the Mediterranean basin (Dupuy et al., 2020; Ruffault et al., 2020; Turco et al., 2018). According to Moreira et al. (2020), burnt areas may be further amplified by land use and management changes that increase fuel load and continuity.

Fire management strategies in the Mediterranean Europe place emphasis on fire suppression which can indeed lead to higher fuel load and fuel connectivity as encapsulated in the term 'firefighting trap' which culminates in hindering suppression under

extreme fire weather, ultimately leading to more severe and usually larger fires (Moreira et al., 2020). Fire management should be enriched, comprising also prevention and adaptation measures (Alcasena et al., 2019; Fernandes et al., 2013). This holistic point of view has been included in the new EU Forest Strategy for 2030 (European Commission, 2021) that explicitly considers fire prevention as an integral component for maintaining and enhancing the resilience of European forests. Further underlining this, in the recent report on wildfires of the United Nations Environmental Program (2022), a radical change in Governments

spending on wildfires was called for, with the aim to rebalance governments' investment from reaction and response to prevention and preparedness.

Seasonal forecasting of weather conditions conductive to fires (fire weather), is of paramount importance for implementing effective fire prevention. The prediction of unfavourable conditions prior to each fire season may support policymakers and civil protection agencies to implement adequate fuel management policies in vulnerable regions, along with optimising fire-

fighting resources to mitigate the adverse effects of forest fires (Turco et al., 2019).

For the relationship between meteorological conditions and fire danger, different indices are used worldwide that assess fire danger for research and operational purposes with the Canadian Fire Weather Index (FWI) being one of the most widely used systems (Field et al., 2015). FWI has been shown to correlate well with fire activity globally (Abatzoglou et al., 2018; Bedia et al., 2015) and regionally, including parts of Europe (e.g., Dupuy et al., 2020; Karali et al., 2014; Ruffault et al., 2020). Since

2007, the FWI has been adopted at the EU level by the European Forest Fire Information System (EFFIS), component of the Copernicus Emergency Management Service (CEMS), to assess fire danger level in a harmonized way throughout Europe after several tests on its validity and robustness for the European domain (San-Miguel-Ayanz, 2012). EFFIS provides short term FWI forecasts, as well as monthly and seasonal forecasts of temperature and rainfall anomalies that are expected to prevail over European and Mediterranean areas for a time window of seven months. To the best of our knowledge, only two studies

so far, have assessed seasonal fire danger predictions for Europe. The first study is by Bedia et al. (2018), in which the authors provided seasonal probabilistic predictions of FWI for Mediterranean Europe by utilizing the ECMWF System-4, focusing on the calibration of model outputs prior to forecast verification, as well as on the analysis of FWI forecast quality compared to reference observed values. In the second study by Costa-Saura et al. (2022), the performance of different seasonal forecasting

systems to predict several indicators relevant to forestry and agriculture for Central Europe and the Mediterranean, including FWI, was assessed.

The current study aims to provide high-resolution probabilistic FWI seasonal forecasts for Attica, Greece, employing the methodology of Bedia et al. (2018) and further expanding it through statistical downscaling. Moreover, it aims to assess the ability of these forecasts to provide robust information and support fire management decisions in the Attica region. Attica encompasses the entire metropolitan area of Athens, the country's capital, and largest city with approximately 3.8 million inhabitants (census of 2021). It is one of the country's most vulnerable regions to rural and peri-urban forest fires due to its complex topography, flammable vegetation, high concentration of population and activities as well as its extensive Wildland-Urban Interface (WUI) (Mitsopoulos et al., 2020; Salvati and Ranalli, 2015).

The catastrophic fires that took place in Attica during the summer of 2021 that burnt more than 150,000 ha (Evelpidou et al., 2022) of forests and arable land underpinned the timeliness and need for this study. These fires broke out during the most severe and the longest heatwave (maximum daily temperature reached 43.9°C, while heatwave conditions prevailed for 10 days) occurred in Attica in the last decades according to the meteorological records of the National Observatory of Athens. Our assessment includes the verification of the FWI ECMWF SEAS5 forecasts against gridded observations using a probabilistic tercile based approach and a qualitative comparison of predicted years with above normal fire danger conditions using historical fire occurrence data.

The paper is organized as follows. In the next section, the data and methods are introduced. In Sec. 3, the results on the forecast performance of FWI, its subcomponents and the input meteorological variables to the FWI system for Attica region are presented, together with the results of the qualitative evaluation of above normal fire danger conditions against historical fire occurrence data. In Sec. 4, the performance of the single meteorological variables, the impact of spin-up and lead time on fire danger forecast performance, as well as the qualitative evaluation of fire danger forecasts, are discussed. Finally, in Sec. 5, the main conclusions and suggestions for future work are discussed.

## 2    Materials and methods

### 2.1    Fire Weather Index (FWI)

FWI is a daily meteorologically based system used worldwide to estimate fire danger in a generalized fuel type-mature pine forest (van Wagner, 1987). According to Wotton (2009), fire danger refers to the assessment of both the static and dynamic factors of the fire environment which determine the ease of ignition, rate of spread, difficulty of control and impact of a fire. The meteorological inputs to the system are daily noon values of air temperature, relative humidity, wind speed and 24-h precipitation (Stocks et al., 1989). The FWI system consists of six subcomponents each measuring a different aspect of fire danger (van Wagner, 1987). The first three primary sub-indices are fuel moisture codes, which are numeric ratings of the moisture content of the forest floor and other dead organic matter. The Fine Fuel Moisture Code (FFMC) is a numeric rating of the moisture content of litter and other cured fine fuels. FFMC is an indicator of the relative ease of ignition and the

flammability of fine fuel, having a fast response to weather variations (approximately 0.5 days under "standard" conditions, i.e., noon temperature 25°C, relative humidity 30% and wind speed 10km/h). The Duff Moisture Code (DMC) is a numeric rating of the average moisture content of loosely compacted organic layers of moderate depth. This code gives an indication of fuel consumption and is characterised by a medium-term response to weather variations (approximately 10 days). The

Drought Code (DC) is a numeric rating of the average moisture content of deep, compact organic layers. DC has a long-term response (about 50 days) to weather variations and is a useful indicator of seasonal drought effects on forest fuels, as well as the amount of smouldering in deep duff layers and large logs. The two intermediate sub-indices, Initial Spread Index (ISI) and Build-Up Index (BUI), are fire behaviour indices. The ISI is a numerical rating of the expected fire rate of spread which combine the effect of wind and FFMC. The BUI is a numerical rating of the total amount of fuel available for combustion that

combines the DMC and the DC. The resulting index is the Fire Weather Index (FWI), which combines ISI and BUI. FWI represents frontal fire intensity (van Wagner, 1987) and can be used as a general index of fire danger (Wotton, 2009). Each component of the FWI System has its own scale, but for all of them a higher value indicates more severe burning conditions (de Groot, 1987). A more analytical description of the FWI system and its subcomponents can be found in van Wagner (1987) and Wotton (2009). The structure of the index and the meteorological variables needed for its calculation are presented in Fig.

A1.

## 2.2   Seasonal forecast data and reference observations

### 2.2.1   ECMWF SEAS5 dataset

In the framework of the current study, the fifth generation ECMWF seasonal forecasting system (SEAS5) (Johnson et al., 2019) available in the C3S Climate Data Store (CDS) (DOI: 10.24381/cds.181d637e) was utilized. SEAS5 has been

operational since November 2017, replacing System 4. The system includes updated versions of the atmospheric (IFS) and ocean (NEMO) models with the addition of the interactive sea-ice model LIM2 (Johnson et al., 2019). The set of re-forecasts (hindcasts) available in the CDS starts on the 1st of every month for the years 1993-2016 and contains 25 ensemble members. The data from these re-forecasts are used to verify the forecasting system and calibrate real-time forecast products. Real time forecasts (from 2017 onwards) consist of a 51-member ensemble initialised every month and integrated for 7 months. The

seasonal forecasts are initialised with atmospheric conditions from ERA-Interim (Dee at al., 2011) until 2016 and the ECMWF Operational Analysis since 2017. Re-forecast and forecast data are available at a global 1x1 degree grid.

For the daily FWI calculations, the SEAS5 instantaneous outputs at 12 UTC for 2-m air temperature, northward and eastward 10-m wind components, 2-m dewpoint temperature, and daily accumulated precipitation were used. The 12 UTC was used as a proxy for local noon values required as input to FWI as proposed by several previous studies for the Mediterranean and

Greece (e.g., Bedia et al., 2012, 2018; Herrera et al., 2013; Papagiannaki et al., 2020). Additionally, according to Papagiannaki et al. (2020), during the fire season the meteorological conditions at 12 UTC (i.e., 15 LST) are highly conductive to the occurrence and spread of fires as corroborated by the Hellenic Fire Service, thus, the respective fire danger predictions are

considered to be particularly useful from an operational perspective. Moreover, relative humidity needed for FWI calculations was computed from air and dew-point temperatures. Concerning precipitation, data correspond to the accumulated values since the initialization time, therefore differences with the previous day's values were computed (de-accumulation) to obtain daily accumulated values for each grid point.

It should be noted that in order to commence the calculations of FWI, default initial values of FFMC, DMC, and DC were used. This means that a spin-up period was required to minimize the effects of errors in the initial conditions used in its calculation. Given that the longest time lag of the fuel moisture codes, as described above, is about 50 days, a spin-up period of up to two months was considered sufficient for both FWI and/or its subcomponents. A fire season spanning from May to September (MJJAS), that coincides with the dry season in Attica according to the records of the Hellenic National Meteorological Service, was considered and six different experimental setups for FWI calculations were implemented. In particular, we performed SEAS5 MJJAS fire danger forecasts initialized in March and April (two months and one month in advance of the target fire season, respectively), without and with spin-up, using both SEAS5 and ERA5-Land data (Figure 1). In the case of spin-up, in 1-month (2-month) lead time forecasts, the FWI time series for April (March and April) were firstly calculated for the index to stabilize and were then removed from the analysis.

### 2.2.2    ERA5-Land reanalysis dataset

As reference observational dataset, the state-of-the-art global reanalysis dataset ERA5-Land (Muñoz-Sabater, 2019) of Copernicus CDS (DOI: 10.24381/cds.e2161bac) was used. ERA5-Land comes with a series of improvements compared to ERA5 making it more accurate for all types of land applications. The dataset provides a total of 50 variables describing the water and energy cycles over land, globally, hourly, and at a spatial resolution of 9 km from 1950 to present (Muñoz-Sabater et al., 2021). To be consistent with the SEAS5 data, 2-m air temperature, 2-m dewpoint temperature, 10-m northward and eastward wind components at 12UTC, and daily accumulated precipitation were used for the calculation of daily FWI values.

### 2.3    Statistical downscaling of seasonal forecasts

To statistically downscale the seasonal forecasts at the ERA5-Land horizontal resolution a two-step approach was followed. In particular, the seasonal forecast meteorological variables used to calculate FWI were initially regridded to the ERA5-Land grid by means of bilinear interpolation and next, bias correction was applied using the empirical quantile mapping (EQM). This two-step approach is the reversed order of the bias correction and spatial disaggregation framework, which has been previously used to statistically downscale global and/or regional models for both climate change and seasonal forecast studies (Lorenz et al., 2021; Markos et al., 2018; Varotsos et al., 2022). Regarding the bias correction method, EQM works by adjusting the 1-99 percentiles of the predicted empirical probability density function (PDF) based on the observed empirical PDF, while for lower or higher values falling outside this range, a constant extrapolation is applied using the correction obtained for the

1st or 99th percentile, respectively. For more information on how EQM works, the reader may refer to the studies of Manzanas

et al. (2018, 2019), Manzanas (2020) and Bedia et al. (2018).

In this study bias correction was applied using daily data for the period May to September using a moving window width of 31 days to adjust the intra-seasonal biases originating from the model's behaviour (i.e., model drift, Manzanas (2020) and references therein). Following Bedia et al. (2018), FWI was bias corrected after its calculation from the regridded fields of temperature, relative humidity, wind speed and precipitation to avoid unrealistic FWI trends that could occur by calculating FWI from the bias corrected meteorological variables. Nevertheless, results of the statistically downscaled temperature, relative humidity, wind speed and precipitation are also presented in the following sections.

## 2.4  Metrics and methodology of fire danger forecast verification

According to WMO (2020), measures of historical predictive skill are an essential component of seasonal forecasts as they provide the users an indication of the trustworthiness of the real-time forecasts. There are many different skill measures describing the quality of specific forecast attributes that are estimated by calculating the corresponding properties of the set (e.g., discrimination, reliability etc.) of hindcasts paired with reference observations (WMO, 2020). In the framework of the current study, the probabilistic Relative Operating Characteristic (ROC) skill score, measuring forecast discrimination, together with the reliability diagrams were used, to assess the potential skill and usefulness of fire danger seasonal forecasts after spatial disaggregation and bias adjustment.

### 2.4.1  ROC skill score (ROCSS)

ROC skill measures the frequency of occasions when the system correctly distinguished between events occurring and not occurring (Jolliffe and Stephenson, 2003). ROC is based on the ratio between the hit rate and the false alarm rate and is evaluated separately for each category (above normal, normal, or below normal). ROC Skill Score (ROCSS) ranges from -1 (perfectly bad discrimination) to 1 (perfectly good discrimination). A value of zero indicates no skill compared to a random prediction or the climatological value.

As in previous studies (e.g., Bedia et al., 2018; Manzanas et al., 2014; Mercado-Bettín et al., 2021), a tercile-based probabilistic approach for forecast verification was applied. In order to assess fire-danger forecast performance, the easyVerification (MeteoSwiss, 2017), SpecsVerification (Siegert, 2020), and VisualizeR (Frías et al., 2018) R packages, were used for skill calculation and visualization. The ROCSS were calculated at each grid-point for the different tercile categories depending on the examined parameter, e.g., the upper tercile for FWI, temperature and wind speed or the lower tercile for relative humidity and precipitation, averaged over the verification period and maps depicting the spatial variations in their skill scores for the different initialization times were constructed.

Moreover, tercile plots for the FWI (and its subcomponents) for Attica were built to complement the spatial analysis provided by the ROCSS maps, presenting the performance of the seasonal forecast along the hindcast period. In order to build a tercile plot for a given variable, the observations along with the bias corrected multi-member ensemble predictions were categorised

into three tercile categories, considering values above (upper tercile), between (middle tercile) or below (lower tercile) the respective climatological values within the 1993–2016 period. Subsequently, a probabilistic forecast was computed year by year considering the number of members falling within each category. Moreover, the observed category according to the ERA5-Land dataset is provided in the plot, to facilitate a visual comparison of hits and misses of the forecast system along the hindcast period.

### 2.4.2 Reliability diagrams

Reliability diagrams are diagnostic tools measuring how closely the forecast probabilities of a specific event (for instance a particular tercile category) correspond to the observed frequency of that event (Weisheimer and Palmer, 2014). According to WMO (2020), in the context of decision-making, forecast reliability plays an important role in making a prior assessment of the benefits of using seasonal forecast information. A construction of a reliability diagram involves binning forecasts by probability category and plotting these values against the observed frequencies (WMO, 2020). For a perfectly reliable forecasting system, the line obtained would match the diagonal (perfect reliability line). The reliability line that best fits the points in the diagram is calculated applying least squares regression weighted by the number of forecasts in each probability bin. Based on the slope of the reliability line and the uncertainty associated with it, six easy-to-interpret categories can be defined: perfect, still very useful, marginally useful+, marginally useful, not useful, and dangerously useless (Manzanas et al., 2018). The marginally useful+ category differentiated those cases for which the reliability line lies within the skill region (Brier Skill Score>0, shaded in grey). The reader can refer to Frías et al. (2018) and Manzanas et al. (2018) for more information on the construction of the reliability diagrams.

It should be noted that concerning FWI (and its subcomponents), both in the tercile maps/plots and the reliability diagrams, only the results of the above normal conditions (upper tercile category) are discussed in the main body of the paper, as high FWI (and its subcomponents) values are related to increased fire danger conditions and, hence, to increased wildfire activity (e.g., Urbieta et al., 2015).

### 2.5 Qualitative evaluation of above normal fire danger conditions against historical fire occurrence

A qualitative evaluation of the ability of FWI hindcasts to predict actual fire occurrence as obtained by historical fire records was performed. To this aim, records of national wildfire time series data for the period between 2000 and 2016 were obtained from the Hellenic Fire Service online database (https://www.fireservice.gr/el_GR/synola-dedomenon). As these data concern both forest and urban fires, only fire events that burnt at least 1ha of forest or forested areas were extracted from the database. Burnt areas less than 1ha were excluded from our analysis, to limit the uncertainties associated with the recording of small fires in fire databases as was reported in previous studies (Jiménez-Ruano et al., 2017; Turco et al., 2013). Regarding the number of fires and the respective burnt areas, these were constrained for the months covering the fire season as defined in the current study (i.e., from May to September). We decided to exclude the fire data for the hindcast years between 1993–1999,

as they were recorded by the Hellenic Forest Service following a different methodology that is not compatible with the Fire Service's one.

Regarding the approach to the qualitative evaluation, the years between 2000 and 2016 were characterized as high fire activity years since the number of fire events for the entire Attica domain for each year was greater than the median of the fire events observed for the whole period. Moreover, only the years with fire danger (based on ERA5-Land) in the upper tercile category (above normal conditions) were selected from the tercile plots for Attica and the relevant proportion of ensemble members predicting upper tercile values was recorded. Consequently, the number of fires per year were shown along with the abovementioned proportion.

## 3    Results

The results section is organized in two parts presenting: a) the forecast performance of the FWI, its subcomponents and the input meteorological variables to the FWI system and b) the qualitative evaluation of above normal fire danger conditions against historical fire occurrence data.

### 3.1    Forecast performance of meteorological variables and fire danger components

The quality of the downscaled fire danger hindcasts for Attica was initially assessed via the ROCSS. In Figure 2, the spatial distribution of the ROCSS for the upper tercile category of the FWI for MJJAS fire season for both lead times, with and without the performance of spin-up, are presented. Statistically significant ROCSS greater than 0.4 were found almost in the entire domain for the 1-month lead time experiments, while higher scores (>0.6) were found for the 2-month lead time experiments. In order to complement this spatial analysis, Figure 3 depicts the tercile plots of FWI averaged over Attica, for both lead time experiments providing a year-to-year visual comparison between hindcast tercile categories and the corresponding observed values as obtained by ERA5-Land. These spatially averaged predictions for the upper tercile of FWI for all experiments indicate statistically significant positive ROCSS resonating the spatial analysis results (Fig. 2). Reaching increasingly higher values, for the 1-month lead time forecasts the ROCSS was 0.45 for no spin-up experiment, 0.57 with spin-up using the SEAS5 model data and 0.62 when the ERA5-Land were implanted in the spin-up procedure. For the 2-month lead time forecasts, higher ROCSSs were calculated for all spin-up experiments compared to 1-month lead (the attained values were 0.66, 0.73 and 0.7, respectively) with the SEAS5 performing slightly better than the observations. Regarding the temporal performance on a year-by-year basis, both lead time experiments depicted high agreement (60-80%) among the members for half of the years with observed above normal conditions.

To further elaborate in the fire danger forecast verification, the reliability diagrams are presented in Figs. A2-A3. The upper tercile FWI predictions for 2-month lead time experiments were classified as perfectly reliable, while predictions fell in marginally useful+ category for 1-month lead time experiments.

Considering the forecasted meteorological variables used in FWI calculations, the ROCSS were calculated for 1-month and 2-month lead time forecasts only when the variable indicates high fire danger conditions, i.e., high air-temperature, low relative humidity, low total precipitation and high wind speed. Thus, the ROCSS for the upper tercile category of air temperature and wind speed, as well as the lower tercile category of relative humidity and total precipitation for the different lead times are presented in Figure 4. Both lead time forecasts of relative humidity and wind speed exhibited high discrimination skills, temperature exhibited low skill almost for the entire domain, while precipitation showed no skill for both experiments. In particular, for relative humidity, statistically significant ROCSSs, greater than 0.6 for 1-month lead time forecast, were attained for the entire domain, while ROCSS ranges between 0.6-0.8 for the 2-month lead time forecast. For wind speed, statistically significant discrimination skill scores between 0.4 and 1.0 were attained for 1-month lead time, while lower values (0.4-0.6) were found for 2-month lead time forecast mainly in the eastern part of the area of interest. Overall, the highest skills averaged over the study domain, were found for the lower tercile of relative humidity (0.73, perfect), the upper tercile of wind (0.45, marginally useful+) and the upper tercile of temperature (0.34, marginally useful+) for 1-month lead time forecasts (not shown).

Given that relative humidity and wind speed demonstrated high discrimination skill for both lead-time experiments, the ROCSSs of the FWI subcomponents that directly depend on these variables were further investigated. In particular, the ROCSSs for Fine Fuel Moisture Code (FFMC) and Duff Moisture Code (DMC) that receive relative humidity as input variable as well as the Initial Spread Index (ISI) which integrates the fuel moisture of fine fuels (FFMC) and near-surface wind speed, were assessed. All fuel moisture subcomponents presented poor discrimination scores (ROCSS<0.3 averaged over the area) for both lead time experiments and depending on the spin-up experiment were classified as not useful or dangerously useless (not shown). The only exceptions are the 1-month lead time FFMC forecast without spin-up and the 2-month lead time DC with spin-up using observations, which were classified as marginally useful+ (not shown). ISI differs as can be seen in Figure 5, showing the spatial distribution of the ROCSS, where statistically significant scores (>0.4) were found almost in the entire domain for both lead time experiments, with higher scores depicted for 1-month lead time. Moreover, the spatial pattern of the ROCSS does not differ within each lead time experiment between the different spin-up experiments. Looking into the tercile plots (Fig. 6), it is evident that the highest ROCSSs for ISI upper tercile predictions are found for the 1-month lead time experiments, having minor differences between the different spin-up experiments (0.85-0.87). Lower values were found for the 2-month lead time experiments, however, the ROCSSs remain high (0.6). From the interannual perspective, concerning 1-month lead time ISI forecasts, most of the observed above normal years were fairly predicted by SEAS5 (by 50%-60% of the members). For 2-month lead time experiments, ISI hindcasts tend to underestimate the observed above normal events, as less than 40% of the above normal years were predicted by most of the members (by more than 60% of the members). Moreover, the FWI and ISI forecast probabilities for 2021 fire season are presented in Figs. 3 and 6. Here, most of the ensemble members (>70%) predict above normal conditions for both FWI and ISI for a year with elevated fire activity, supporting the case of their usefulness for providing fire danger forecasts under operational usage. Lastly, according to the reliability diagrams, the ISI

285     predictions for 1-month lead time experiments are classified as perfectly reliable, while 2-month lead time experiments fall in the marginally useful+ category (Fig. A4-A5).

## 3.2     FWI and ISI predictions against fire occurrence

In this section, the focus will only be on FWI and ISI as these were found to perform better with respect to their ROCSSs and respective reliability. The qualitative evaluation of above normal fire danger conditions against historical fire occurrence was thus implemented for the FWI and ISI subcomponent, for both lead times and only for the spin-up experiments with the highest discrimination scores, as discussed in the previous section.

295     In order to decide which fire occurrence aspect should be considered, the correlation between FWI and ISI hindcasts with burnt areas and the number of fires for the years 2000–2016 was calculated and revealed moderate correlation between FWI and ISI with the number of fires (r=0.55 and 0.45 respectively, p-value<0.05) and no statistically significant correlation with burnt areas. Similar results were reported in a recent study of Galizia et al. (2021) suggesting that fire-prone pyro-regions, with Greece and Attica categorized as such, present moderate (>0.4) and strong (>0.6) positive correlations of the number of fires 300     with the FWI and ISI, respectively. Thus, the number of fires instead of burnt area was eventually favoured as the variable of choice for the qualitative evaluation of fire danger hindcasts.

Figure 7 depicts the number of fires (with burnt area greater than 1ha) per year, for the years between 2000 and 2016 of the hindcast period and the respective proportion of ensemble members predicting above normal FWI and ISI values as obtained by the tercile plots averaged over the entire Attica domain (Figs. 3 and 6).Concerning both FWI and ISI, the prediction of years 305     with increased fire activity (i.e., the years with total number of fires greater than the 2000–2016 median based on the fire records), was clearly dependent on the lead time of the forecasts. It is reminded that only the years with observed (based on ERA5-Land) fire danger in the upper tercile category (above normal conditions) were taken into account. This includes also the 2003, 2009 and 2010 high fire activity years which according to the ERA5-Land observations fall in the middle (2003, 2009) and lower (2010) terciles.

310     As seen in Fig. 7, half of the remaining years with increased fire activity are indeed captured by more than 60% of the ensemble members by at least one of the 1-month or 2-month lead time FWI forecasts. The high fire activity of 2007 is captured only by 2-month lead time experiment, while 2012 is missed by both lead time experiments. Moreover, 2016 is overshot by the 1-month lead time experiment. Regarding ISI, more than half of the years are captured with the percentage of ensemble members varying between 50-80% by at least one of the lead time experiments. Lastly, the high fire activity of the 2000 and 2012 fire 315     seasons are not captured by 2-month lead time forecasts.

## 4 Discussion

### 4.1 Prediction skill of single meteorological variables

Before delving into the indicators more directly associated with fire danger, the individual meteorological variables that serve as input to the FWI system were examined. As discussed in Sect. 3.1, it appears that relative humidity and wind speed exhibit high discrimination skills, temperature exhibits low skill, while precipitation showed no skill for both lead time experiments. For all meteorological variables, the forecast performance declines as the forecast lead time (i.e., the period between the target fire season and the initialization date of the forecast) increases, which is in line with previous studies (e.g., Doblas-Reyes et al., 2013; van den Hurk et al., 2012).

Looking at the individual variables, according to Mishra et al. (2019), limited predictive skill of seasonal temperature and very low skill of seasonal precipitation was found over entire Europe based on the EUROSIP multi-model framework, including the ECMWF System 4, the predecessor of SEAS5. Regarding the area under study, it is part of the Mediterranean region which is considered an area of transition between subtropical and mid-latitudes, where seasonal forecasts are challenging, therefore, the assessment of the added value and the identification of limitations of seasonal forecast products are of paramount importance when developing climate services (Calì Quaglia et al., 2022). The same study found statistically significant temperature anomaly correlations over the eastern Mediterranean between the SEAS5 and the ERA5 reference dataset, however, summer ROC skill score was not discussed in that study. Additionally, summer precipitation showed limited skill, located mainly at the western part of the Mediterranean. In general, the climate of the western Mediterranean is more predictable than the eastern part of the domain, probably due to the influence of El Niño-Southern Oscillation (ENSO) and North Atlantic Oscillation (NAO) teleconnections (Calì Quaglia et al., 2022; Frías et al., 2010). Concerning relative humidity, our results are in line with previous studies (Bedia et al., 2018; Bett et al., 2018) who found significant skills over the eastern Mediterranean using the ECMWF System 4 forecasting system. Finally, wind speed can be considered a promising variable regarding skill, as it is more closely related to the larger-scale atmospheric circulation than more complex processes like precipitation. According to Bett et al. (2022), the wind skill was found to be patchy throughout Europe especially during summer using however the Sys4 forecasting system. The high wind skill for Attica empowers the discussion of the next section as the FWI is highly sensitive to wind speed (Karali et al., 2014). In addition, Kassomenos (2010) found that the Etesians (dry north winds prevailing during summer) are very often associated with the development of extreme wildfires in Greece, while Pashalidou and Kassomenos (2016) pointed out that mesoscale and local systems can play an important role on fire development, as they interact with, and may exacerbate the larger scale circulation patterns.

### 4.2 Impact of lead-time and spin-up on fire danger forecast performance

Concerning the performance of the FWI and its subcomponents, according to the results presented in Sect. 3.1, it appears that the lead time of the forecast highly affects their skill scores and especially those of FWI and ISI which attained the highest ROCSSs and are therefore discussed here. As far as ISI is concerned, the highest ROCSSs were found for the 1-month lead

time forecasts and can be attributed to the high performance of wind speed for this exact lead time experiment. ISI ROCSSs (Fig. 5) also indicate that the specific subcomponent is insensitive to spin-up as it remains unaffected between experiments and its performance is mostly controlled by the skill of the meteorological variables used for its calculations for the different lead time forecasts. This can be attributed to the fact that ISI is calculated by solely combining wind speed with FFMC, the latter having a fast response (less than one day) to weather variations as presented in Sect. 2.1.

Concerning FWI, its high complexity and the non-linear relationships between its input meteorological variables and subcomponents, makes it difficult to attribute its performance to a single variable and/or subcomponent. According to the results, FWI performs better in 2-month lead time experiments (even in the no spin-up experiment), even though the forecast performance of the single variables as discussed in Sect. 4.1 is decreased. A potential reason could be the higher scores of DC and, therefore, of BUI subcomponent (Fig. A6), compared to 1-month lead, which may be attributed to the memory of the DC subcomponent associated to soil moisture. The improved BUI of 2-month lead time (Fig. A6), combined with the relatively high ISI (Fig. 5) skill scores lead to high FWI performance. The spin-up impact on FWI, which can be seen in Fig. 3, is positive as higher discrimination scores were achieved in both 1-month and 2-month lead time experiments, without however altering the reliability class. From the terciles plots, it is also evident that although spin-up alters the discrimination skill of FWI forecasts, the choice between model or observations in the spin-up procedure plays a minor role.

In summary, depending on the lead time of the forecasts, both FWI and ISI were found useful tools in decision making for the region under study as the scores imply. As several subcomponents of FWI system (such as ISI, BUI, FFMC) can be used by fire management authorities (Wotton, 2009), further research could be directed to utilizing multi-model ensembles in order to study potential improvements in the scores of FWI and these subcomponents.

### 4.3    Qualitative evaluation of fire danger forecasts to predict fire occurrence based on fire statistics

The qualitative evaluation of the best performing forecast experiments of ISI and FWI has been carried out against fire occurrence data presented in Sect. 3.2. Both FWI and ISI forecasts managed to capture high fire activity years adequately (for the period 2000-2016), while the forecast probabilities were found to be highly dependent on the lead time. For half of the high fire activity years, both indices managed to capture high fire activity fire seasons with forecast probabilities greater than 0.6 (>60% of the ensemble members for both lead time experiments). This implies that at least for high activity seasons, the seasonal approach for these two indices can be useful for complementing current fire management tools.

Regarding the misses discussed in Section 3.2, the reason is twofold. On one hand it should be considered that fire activity is not only driven by climate but rather by interactions among climate, vegetation and human activities (Galizia et al. 2021). Thus, a climate-only approach, as proposed here may be proven insufficient for certain years. Disasters such as forest fires arise from a complex interplay between hazard, vulnerability and exposure (GIZ, 2017; IPCC, 2022). The integration of seasonal forecast information (i.e., constituting the hazard component of risk) with other types of information, describing the natural and human capital as well as the vulnerability of the exposed system (Bacciu et al., 2021), is critical in order to enhance planning and decision-making regarding fire prevention and preparedness. On the other hand, the misses highlight the

sensitivity of the results to the ERA5-Land dataset which was used to statistically downscale and evaluate the seasonal forecasts output (Herrera et al., 2019; Mavromatis and Voulanas, 2021). Therefore, further research is needed to investigate the impact of the selected reference dataset on the statistically downscaled forecasts.

## 5    Conclusions

As climate plays an important role in fire dynamics and climate change is increasing the frequency and severity of fire weather, the ability to forecast fire danger conditions prior to the beginning of the fire season can enhance preparedness and support decision making in fire management for fire-prone areas. Moreover, the resilience of the forestry sector may be enhanced by developing dedicated climate services, such as fire danger seasonal forecasts, in order to reduce risks and offer opportunities for long-term reduction of wildfire disasters. The aim of this study is to provide high resolution probabilistic seasonal fire

danger forecasts, utilizing Fire Weather Index (FWI) for Attica Greece and verify these forecasts using probabilistic verification measures for skill assessment (ROC skill score, reliability diagrams). The ultimate goal is to explore whether these forecasts can support disaster management and relevant regional authorities by incorporating such fire risk assessment indicators, in prevention and preparedness plans (Oom et al., 2022).The analysis focuses on the predictability of above-normal (upper tercile) FWI years which have been associated in several studies with increased fire occurrence. Moreover, the study

tried to assess the ability of fire danger forecasts to capture years with increased fire activity, by comparing hindcast years of above normal fire danger conditions with historical fire occurrence data obtained by the Hellenic Fire Service. Our results suggest that depending on the lead time of the forecast, both FWI and ISI present statistically significant high discrimination scores and can be considered reliable in predicting above normal fire danger conditions. Therefore, they can be viewed as valuable climate related alarms of increased fire danger and fire occurrence and may be further exploited by regional authorities

in fire management regarding prevention, preparedness and resources allocation in the Attica region and other fire prone regions and sub-regions in the Mediterranean.

Future work should focus on the assessment of large ensemble approaches utilizing different forecasting systems available in Copernicus CDS, as well as alternative pathways to enhance the skill of seasonal fire danger predictions to be applicable to the whole Greek or even Mediterranean wide domain. Finally, the impact of the selected reference dataset, here ERA5-Land,

on the statistically downscaled forecasts should also be explored.

**Data availability**

The post-processed datasets generated during the current study can be available by the corresponding author on reasonable request.


## Author contributions

AK, KVV conceptualized and developed the methodology; AK, KVV performed formal analysis; AK, KVV, MH wrote the original manuscript draft; CG resources and funding acquisition; AK, KVV, MH, CG and PPN wrote, reviewed, and edited the final manuscript.


## Competing interests

The authors declare that they have no conflict of interest.

## Acknowledgements

The authors would like to thank the Copernicus Climate Change Service for making freely available SEAS5 seasonal forecast and ERA5-Land datasets.

## Financial Support

The authors would like to acknowledge funding from the C3S European Tourism (C3S_422_Lot2_TEC) as well as the H2020
FIRE-RES (grant agreement No 101037419) projects.

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

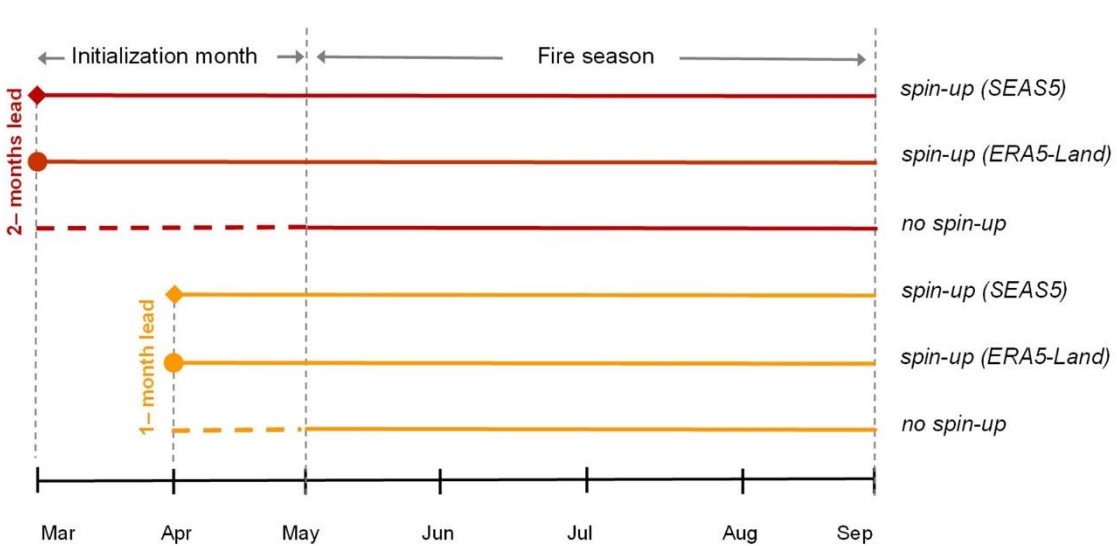

**Figure 1: Experimental setups used for FWI calculations. Forecasts are initialized in April (1-month lead time, in yellow) and March (2-month lead time, in red) while three different experiments concerning the spin-up period (a) with no spin-up (dashed line), (b) with spin-up implanting the ERA5-Land (solid line-circle symbol) and (c) with spin-up using the SEAS5 model data (solid line-diamond symbol) are shown.**


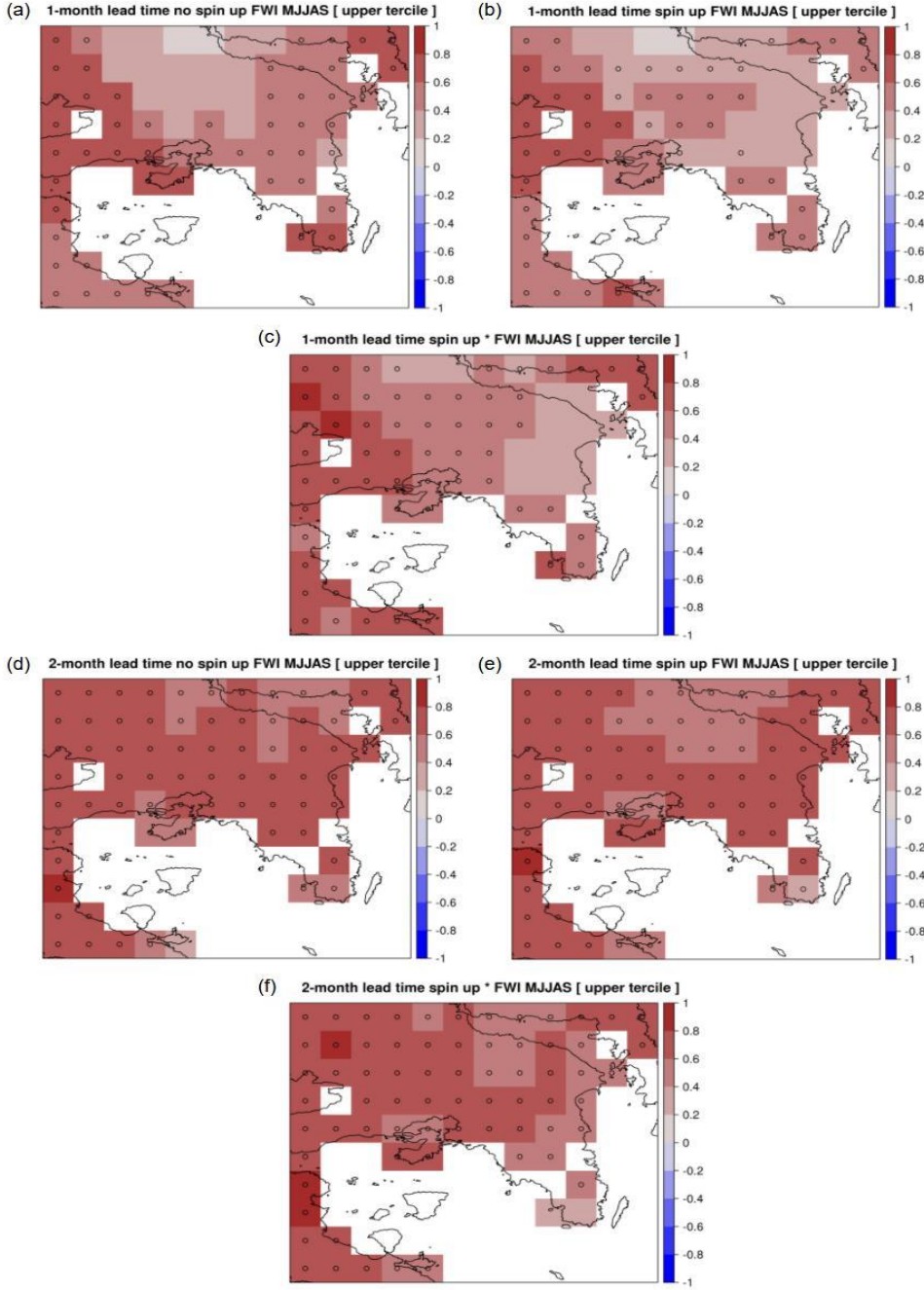

**Figure 2: ROC Skill Scores (ROCSSs) of the upper tercile SEAS5 FWI predictions for 1-month lead time: (a) with no spin up, b) with spin-up using the SEAS5 data, c) with spin-up implanting the ERA5-Land data and 2-month lead time: d) with no spin-up, e) with spin-up using the SEAS5 data and f) with spin-up implanting the ERA5-Land data. The grid points with significant ROCSS values are indicated by circles (α=0.05).**

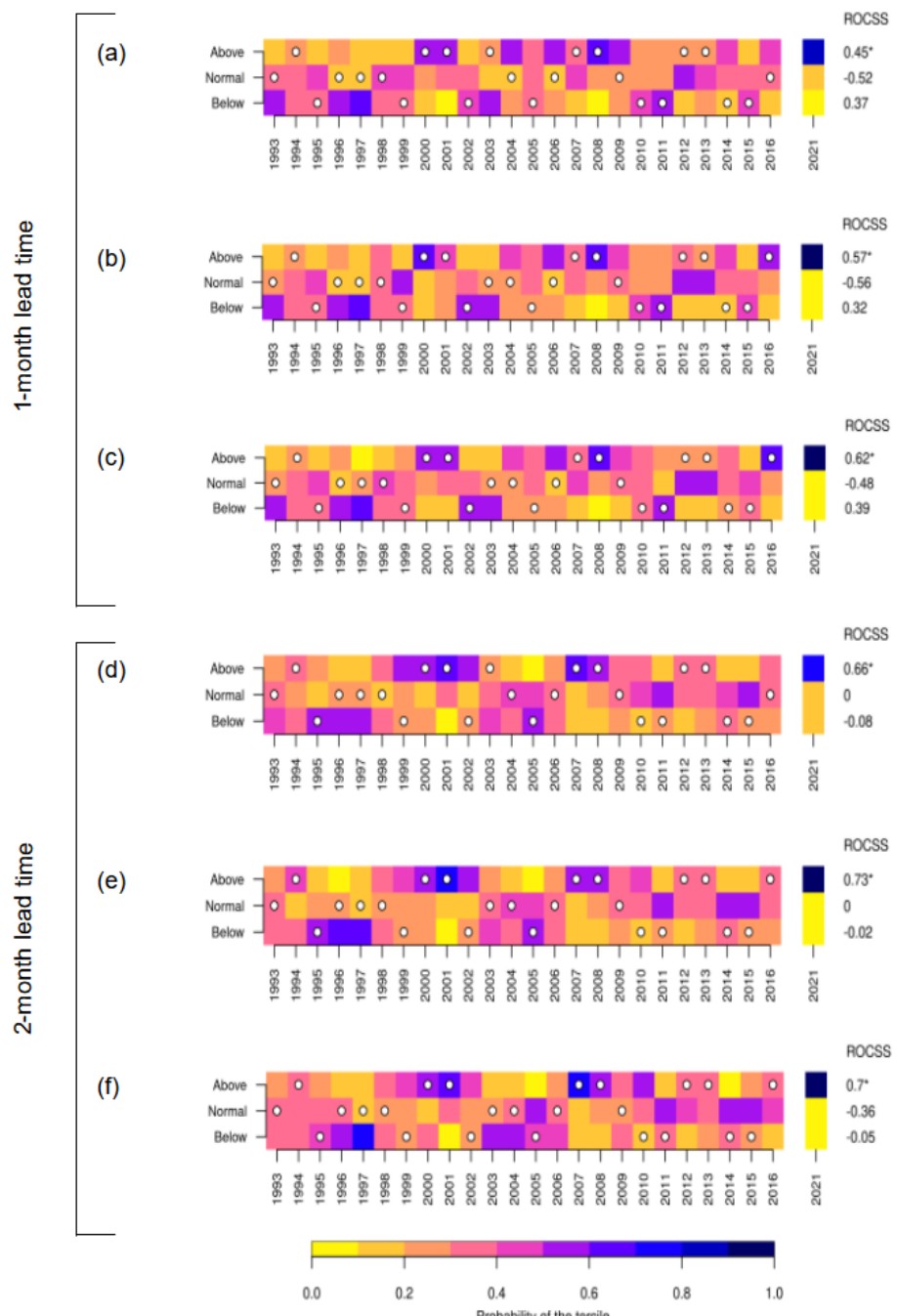

**Figure 3: Tercile plots for May to July FWI predictions covering the hindcast period (1993-2016) for 1-month lead time: (a) with no spin-up, (b) with spin-up using the SEAS5 model data, (c) with spin-up implanting the ERA5-Land data and for 2-month lead time (d) with no spin-up, (e) with spin-up using the SEAS5 model data and (f) with spin-up implanting the ERA5-Land data. Forecast probabilities for the three tercile categories are codified in a yellow (0, no member forecasts in one category) to blue (1, all the**

**members in the same category) scale. The white bullets represent the observed category according to the ERA5-Land dataset. ROCSS values obtained from the hindcast period are shown on the right side of each category and the asterisk indicates significant values (α=0.05).**


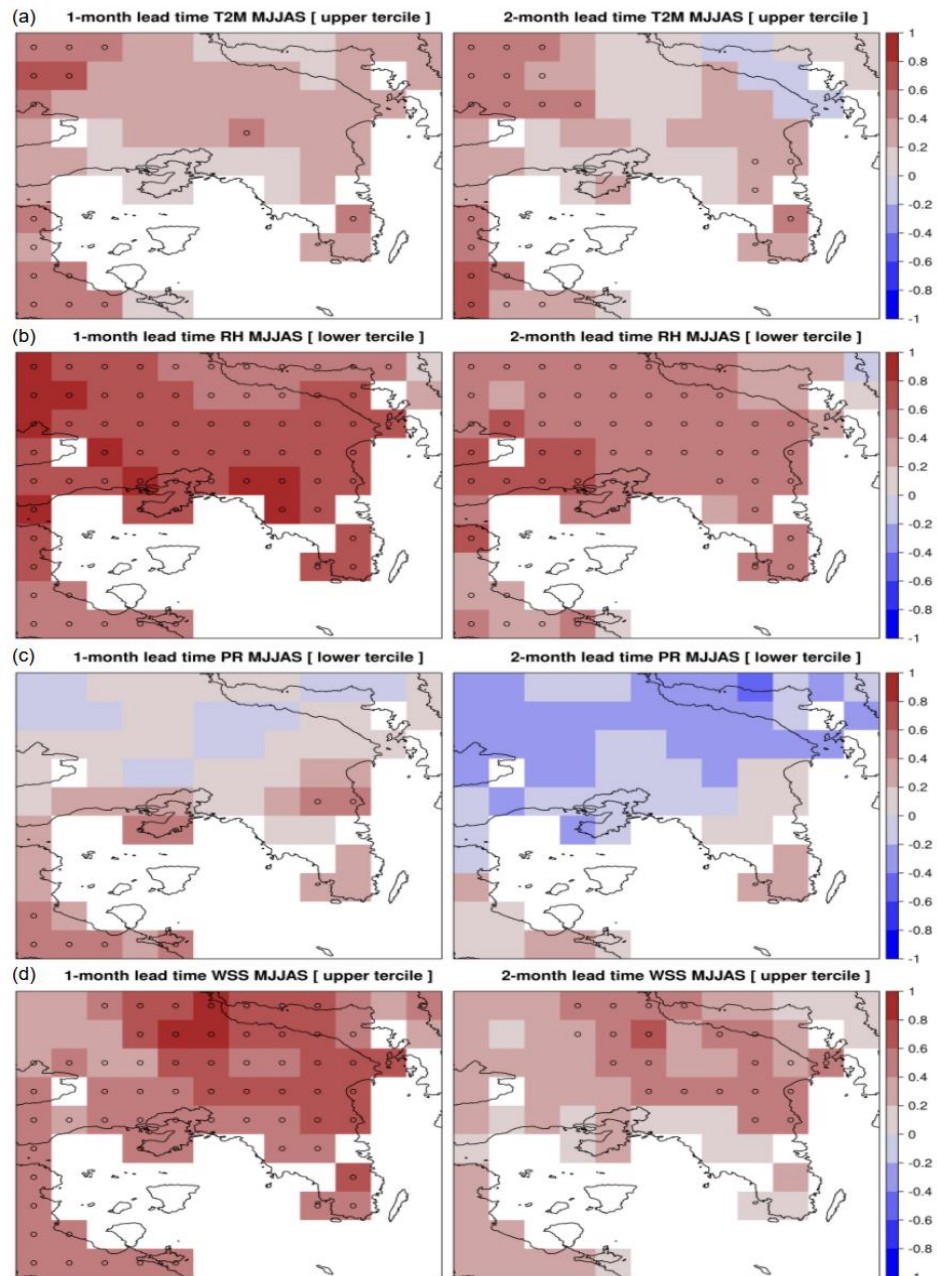

**Figure 4: ROCSSs of the FWI input variables for 1-month (left column) and 2-month lead (right column) time forecasts that correspond to high fire danger values: (a) upper tercile of air temperature (T2M), (b) lower tercile of air relative humidity (RH), (c) (c) lower tercile of total precipitation (PR) and (d) upper tercile of wind speed (WSS). The grid points with significant ROCSS values are indicated by circles (α=0.05).**


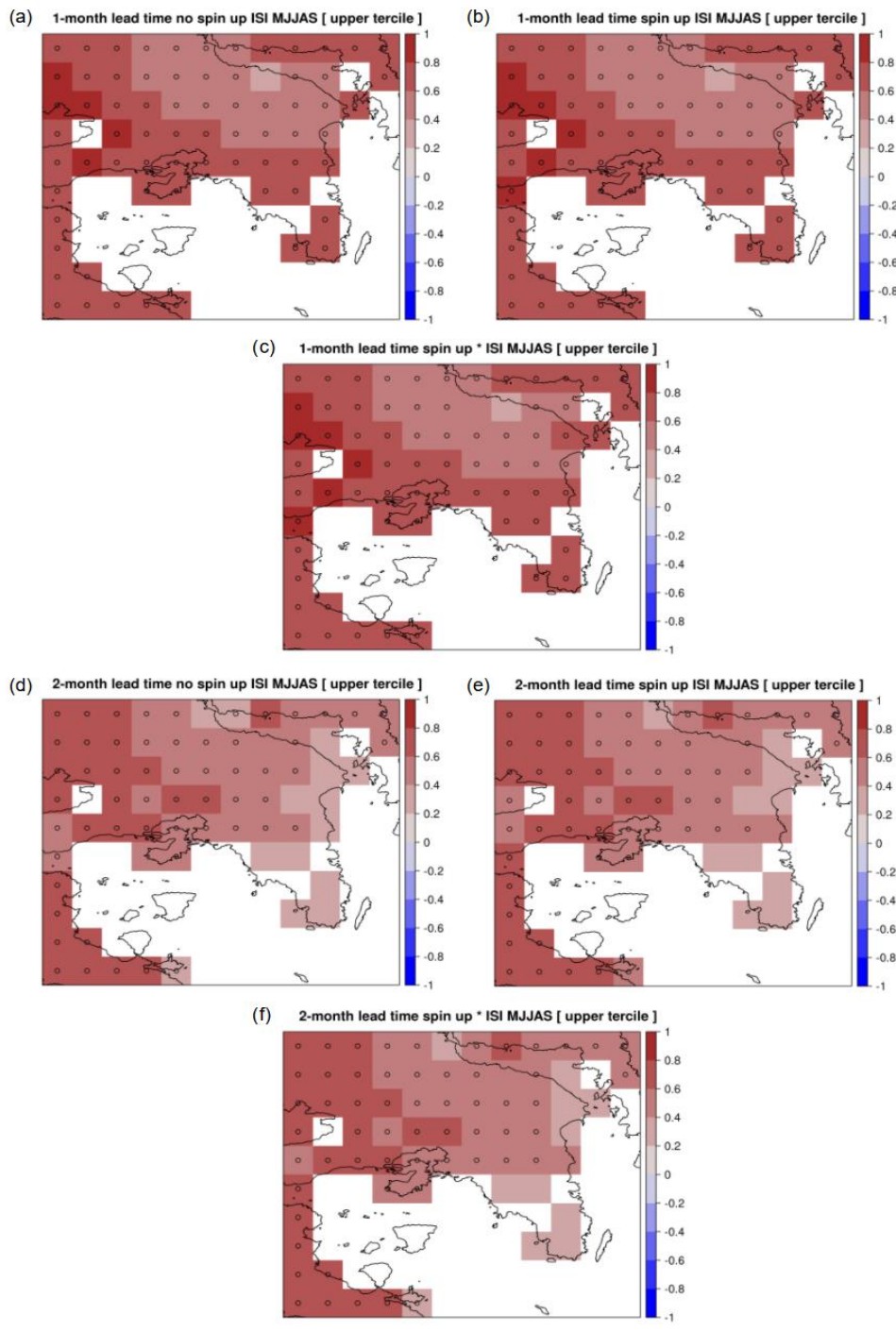

**Figure 5: Same as Fig. 2 but for ISI predictions.**

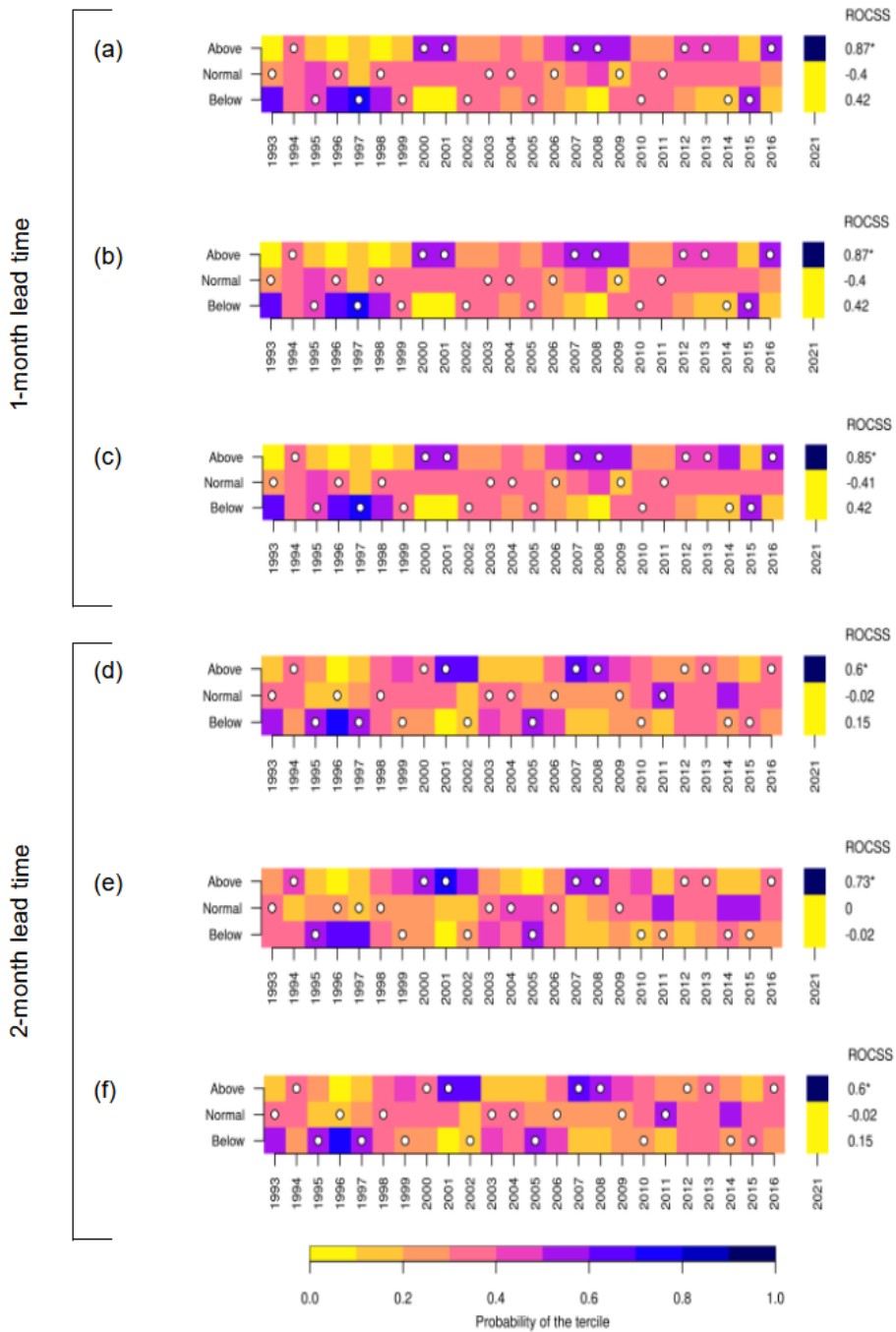


**Figure 6: Same as Fig. 3 but for ISI predictions.**

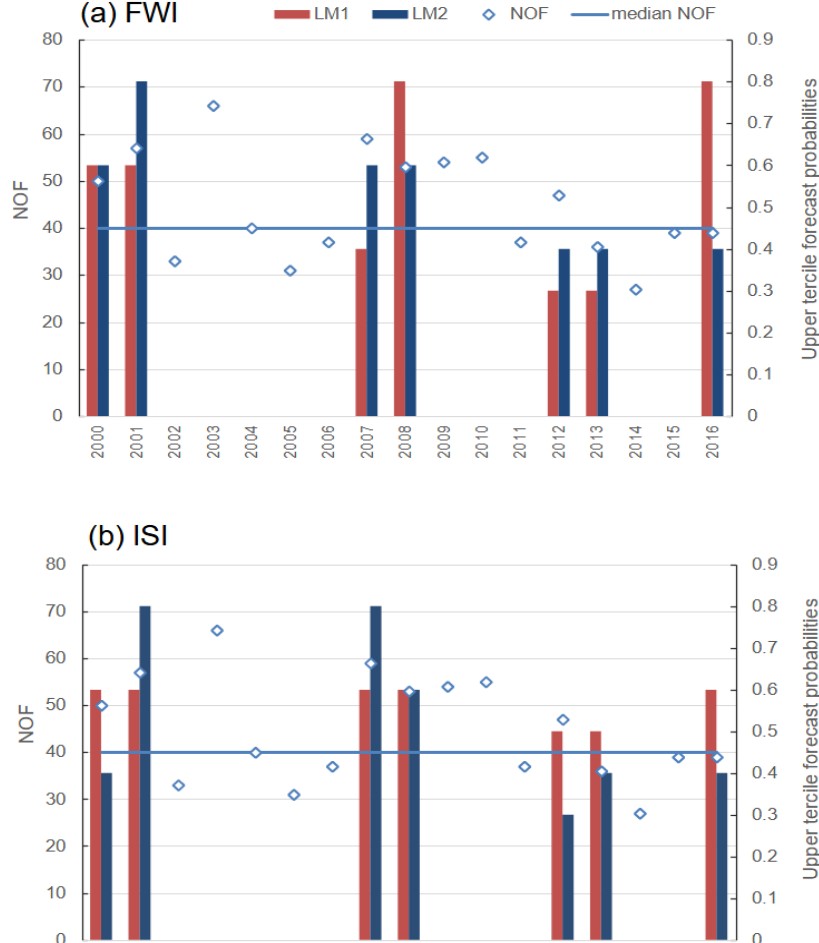


**Figure 7: Annual number of fires (NOF) in Attica (blue diamonds), median of fire events for 2000-2016 (blue line) and upper tercile forecast probabilities of: (a) FWI and (b) ISI for 1-month (red columns) and 2-month (blue columns) lead time best performing spin-up experiments.**
