# Peer review of "Seasonal fire danger forecasts for supporting fire prevention management in an eastern Mediterranean environment: the case study of Attica, Greece"

_Natural Hazards and Earth System Sciences, 2022_

## Author Comment (AC1)

This work aims at providing high resolution probabilistic seasonal FWI forecasts for Attica (Greece) and verifying these forecasts using probabilistic verification measures for skill assessment (ROC skill score and reliability diagrams). To accomplish that, the authors compute FWI and its components for the fire season MJJAS using the dynamic forecasting system SEAS5 with lead times of 0 and 1 month (issued in May and April, respectively). The manuscript describes innovative research with potential operationality in a country that has been recently affected by large wildfires. The main issue with the manuscript is the almost non-existent discussion. Indeed, the majority of the problems found in the manuscript can be solved with the disentanglement of the section "Results and Discussion" into two sections. Furthermore, the paper has many typos and sentences that are not well written. I urge the authors to carefully re-read everything. Thus, since some of the revisions may take some time to tackle, I suggest major revisions.

Comments:

1. Line 96: You say that "FWI represents the frontal fire intensity…". I do not agree with this sentence. Frontal fire intensity is the energy output rate per unit length of fire front. FWI is not that. Can you please elaborate on what you intended to say? Also, I think the explanation of the FWI components is very loose, especially because you introduce FFMC, DMC, and DC explaining what they are but never defining what the abbreviatures mean. Indeed, lines 88 – 91 are not reader-friendly. Moreover, in line 92 when you introduce ISI and BUI for the first time you should put in parenthesis "(Initial Spread Index)" and "(Build-up Index)". You only do this on the second appearance.

Answer:

You say that "FWI represents the frontal fire intensity…". I do not agree with this sentence. Frontal fire intensity is the energy output rate per unit length of fire front. FWI is not that. Can you please elaborate on what you intended to say?

The definition used in the manuscript is based on van Wagner (1987) who stated that FWI represents the intensity of the spreading fire as energy output rate per unit length of fire front (please refer to pp. 3 of the above-mentioned publication). Moreover, according to Wotton (2009), FWI is analogous to Byram's fireline intensity (Byram 1959) and it provides a general summary of fire weather and fuel moisture in a region when a single indicator of general fire potential is needed, for instance when communicating fire danger to the general public.

Therefore, in the revised manuscript the FWI definition will be complemented with the above-mentioned references as follows:

*FWI represents frontal fire intensity (van Wagner, 1987) and can be used as a general index of fire danger (Wotton, 2009).*

Also, I think the explanation of the FWI components is very loose, especially because you introduce FFMC, DMC, and DC explaining what they are but never defining what the

**abbreviatures mean. Indeed, lines 88 – 91 are not reader-friendly. Moreover, in line 92 when you introduce ISI and BUI for the first time you should put in parenthesis "(Initial Spread Index)" and "(Build-up Index)". You only do this on the second appearance.**

A tighter as well as analytical description (mainly for the three moisture sub-indices) of each FWI subcomponent will be provided together with the explanation of the different abbreviations at their first appearance in the revised manuscript as follows:

*The FWI system consists of six components each measuring a different aspect of fire danger (van Wagner, 1987). The first three primary sub-indices are fuel moisture codes, which are numeric ratings of the moisture content of the forest floor and other dead organic matter. The Fine Fuel Moisture Code (FFMC) is a numeric rating of the moisture content of litter and other cured fine fuels. FFMC is an indicator of the relative ease of ignition and the flammability of fine fuel, having a fast response to weather variations (approximately 16 hours under "standard" conditions, i.e., noon temperature 21.1°C, relative humidity 45% and wind speed 12km/h). The Duff Moisture Code (DMC) is a numeric rating of the average moisture content of loosely compacted organic layers of moderate depth. This code gives an indication of fuel consumption and is characterised by a medium-term response to weather variations (10–12 days). The Drought Code (DC) is a numeric rating of the average moisture content of deep, compact organic layers. DC has a long-term response (about 50 days) to weather variations and is a useful indicator of seasonal drought effects on forest fuels, as well as the amount of smoldering in deep duff layers and large logs.*

*The two intermediate sub-indices, Initial Spread Index (ISI) and Build-Up Index (BUI), are fire behaviour indices. The ISI is a numerical rating of the expected fire rate of spread which combine the effect of wind and FFMC. The BUI is a numerical rating of the total amount of fuel available for combustion that combines the DMC and the DC.*

**2. Line 113: Indeed, FWI is to be computed at local time noon. I searched for Greek time, and in summer there is a difference of three hours to 12UTC, i.e., you are computing FWI at 15 local time. As I understand SEAS5 sub-daily values are available in a 6-hour window, which means you had to choose between using values at 9 or 15 local time. It is of paramount importance that you explain this here. Also, your discussion section must take this issue in consideration.**

**Answer:**

FWI is computed using meteorological records at ''noon local standard time" (Stocks et al, 1989). However, when using global long-range forecasts and climate models the appropriate temporal resolution for the input variables to the FWI is not always available therefore, proxies of noon local time are usually used. The choice of 12UTC as a proxy for noon conditions in the Mediterranean and Greece has been proposed in several studies before (e.g., Bedia et al., 2012, 2014, 2018; Herrera et al., 2013, Papagiannaki et al. 2020), therefore it seems a reasonable choice for our study too.

Moreover, according to van Wagner et al. (1987), FWI represents fire danger potential at its midafternoon peak (generally specified as 1600 hours). This is due to the fact that the noon variables in the original FWI System were correlated with fine fuel moisture and test fire data taken later in the afternoon. Finally, according to Papagiannaki et al. (2020), the meteorological conditions at 15 LST are highly conductive for the occurrence and spread of fires and the respective fire danger predictions are considered to be particular useful from an operational perspective based on the experience of the officers of the Hellenic Fire Service. Taking all the above into consideration, 15 LST instead of 9 LST input meteorological data are considered more appropriate to calculate and depict the noon /midafternoon peak of the index and fire activity in the area of interest.

3. **Results and Discussion: Sections 3.1 and 3.2 are mainly a description of the results with a somewhat poor discussion. I recommend the authors to disentangle this section into two separate sections (3 Results; keeping 3.1 and 3.2; and 4 Discussion; 5 Conclusions). Moreover, the authors show a total of 9 figures in the results (and 1 in the methods), which seems too much for a 2-page section (results and discussion). I believe that with a stronger discussion section the 9 figures can better fit the amount of text. However, I recommend the authors to verify if all the figures are essential to the main body of the manuscript (if not, send some of them to supplementary material). For example, Figure 1 is not essential. Above I recommended the authors to design a scheme, which may also be present in supplementary material (for readers who are new to the field understand the methodology of forecasts and lead-times).**

**Answer:** As suggested by the reviewer, in the revised manuscript two separate sections for Results and Discussion will be provided. In the *Results* section the structure will remain the same as in the first manuscript (keeping 3.1 and 3.2), while in the *Discussion* section the following will be discussed: a) the prediction skill of the single meteorological variables used in the FWI system, b) the impact of lead-time and spin-up on fire danger forecast performance and c) the qualitative evaluation of fire danger forecasts to predict fire occurrence based on fire statistics. Part of the discussion is provided in Answer 5 below. Moreover, supplementary material will be provided with some of the figures that are complementary in our analysis (such as Fig. 1, Figs. 8-9 of the first manuscript). Finally, a new figure presenting the different lead time and spin-up experiments will be added in the methodology section. Please see also the answer to comment 10 below.

4. **Lines 280 – 285: It is crucial to understand that FWI is a fire risk index, which only gives a picture of how susceptible a region is to burn. Without an ignition and vegetation prone to burn there will be no fire. It is interesting to see that it is possible to relate FWI forecasts for MJJAS fire season issued in April and May with the number of fires, but I do not think this is a key issue. However, doing such an analysis, why didn't the authors choose Fire Radiative Power (FRP), which could give an aggregated vision of the fire**

**intensity in the region for the period of study? Wouldn't it be a more interesting variable than the number of fires?**

**Answer:**

As mentioned in Wotton (2009) FWI can be used as a general fire danger indicator.  As it relies only on weather forcings, the index describes dangerous weather conditions conductive to uncontrollable fires in a region and not the susceptibility of a given region to forest fires which is highly dependent on other factors such as fuel load, fuel type and connectivity, topography etc. Thus, high fire danger can still be recorded where fires are otherwise inhibited due to e.g., fuel availability or the absence of an ignition.

Despite this limitation, FWI and its subcomponents have already been shown in several studies, to correlate well with fire activity data. In the recent study of Galizia et al. (2021) for European fire regimes, statistically significant moderate/strong correlations were found between FWI/ISI and the number of fires for fire-prone pyro-regions in Europe, with Greece (and Attica) categorized as such. Focusing on Greece, Dimitrakopoulos et al. (2011) found strong correlations between FWI and fire occurrence data, while Karali et al. (2014) found strong relationships between the number of fires and region-specific fire occurrence FWI thresholds.

Moreover, we are aware that recent studies (e.g., Durao et al., 2022) have used FRP/FRE values in order to assess fire intensity. However, our goal in this part of the analysis, is to provide an evaluation of the ability of FWI hindcasts to predict actual fire occurrence and not fire intensity in Attica region, given the strong correlations between fire events and FWI (and its subcomponents) found in previous studies and described above.

5. **Figures 3 – 4: More discussion on these figures is needed. The authors need to develop on why fire season wind speed is better forecasted in March than in April. Why does precipitation show no skill? Also, it is very hard to relate the colours to values. I recommend you use one colour in steps of 0.2 and try to use more contrasting tones of reds and blues (this applies to Figures 2 – 6).**

**Answer:**

In Figure R1 below, the new maps depicting the ROCSS for the upper tercile category of air temperature and wind speed, as well as the lower tercile category of relative humidity and total precipitation for the 1-month and 2-month lead times are presented. After the reviewer's comments the colorbar/colorstep has been changed.

From Fig. R1 is evident that forecasts of relative humidity and wind speed (mainly for 1-month lead time) exhibit high discrimination skills, temperature exhibits low skill almost for the entire domain of study, while precipitation showed no skill for both lead time experiments. It is also evident that the forecast performance declines as the forecast lead time (i.e., the period between the target fire season and the initialization date of the forecast) increases. This is in line with previous studies by Doblas-Reyes et al. (2013) and van den Hurk et al. (2012).

According to our literature search, the Mediterranean region is an area of transition representative of the mid-latitudes, where seasonal forecasts are challenging, therefore the assessment of the added value and the limitations of these products is of paramount importance when developing climate services (Quaglia et al., 2022). According to Mishra et al. (2019) limited predictive skill of seasonal temperature and very low skill of seasonal precipitation is found over Europe based on the EUROSIP multi-model framework, including the ECMWF System 4, the predecessor of SEAS5 used in our study. A recent study for the Mediterranean (Quaglia et al., 2022), found significant temperature anomaly correlations over Eastern Mediterranean between the SEAS5 and the ERA5 reference dataset, however summer ROC skill score was not discussed in the study. According to the same study, summer precipitation showed limited skill, focused mainly on the western part of the Mediterranean. Concerning relative humidity, our results are in line with previous studies (Bedia et al., 2018; Bett et al., 2018) who found significant skills over the eastern Mediterranean using the ECMWF Sys4 forecasting system. Finally, wind speed can be considered a promising area for skill, as it is more closely related to the larger-scale atmospheric circulation than more complex processes like precipitation. According to Bett et al. (2018), the wind skill was found to be patchy throughout Europe especially during summer using Sys4 forecasting system. Focusing on Attica where our results present high skill regarding the wind, Kassomenos (2010) found the Etesians (dry north winds prevailing during summer) to be very often associated with the development of extreme wildfires in Greece, while Pashalidou et al. (2016), pointed out that mesoscale and local systems can also play an important role on fire development, as they interact with, and may exacerbate the larger scale circulation patterns.

The above will be included in the discussion part of the revised manuscript.

[Figure]

Figure R1: ROC Skill Scores of the FWI input variables for 1-month and 2-month lead time MJJAS forecasts that correspond to high fire danger values, i.e., upper tercile of air temperature

(T2M), upper tercile of wind speed (WSS), lower tercile of air relative humidity (RH) and lower tercile of total precipitation (PR). The grid points with significant ROCSS values are indicated by circles (α=0.05).

**6.  Line 27: "as regards" to "regarding".**

**Answer:** "As regards" will be replaced with "regarding" to the revised manuscript following the reviewer's comment.

**7.  Line 84: When defining FWI you should add the Van Wagner reference.**

**Answer:** According to the reviewer's suggestion the reference to van Wagner (1987) will be added in FWI definition (please see also the answer to the 1st comment).

**8.  Line 116: replace "temperature" by "temperatures".**

**Answer:** "Temperature" will be replaced with "temperatures" in the revised manuscript.

**9.  Line 117: Why May – September as fire season? Why do you include May and ignore October? Is this common for Greece?**

**Answer:** It is true that according to the Hellenic Fire Service the official fire season for the whole Greece is the period from May to October.  However, in the current study which focuses on the Attica region, in order to define the fire season we selected the region's particular dry season, which, according to the climatological records of the Hellenic National Meteorological Service (HNMS), covers the months from May to September as can be seen in the following figure (Fig. R2).

[Figure]

Figure R2: Monthly mean precipitation (red line) and number of days with precipitation /wet days (blue columns) for the period 1955-2010 for three indicative stations in Attica region.

**10. Line 118: Why do you stop at 1-month lead time? Wouldn't it be interesting to see the predictive skill of forecasts starting in March? I believe 6-month lead times (7 months prior to target) are available, which means that for September there are forecasts initialized in March. Does it have to do with the spin-up period? Moreover, please try to better explain what a spin-up period is.**

**Answer:**

As has already been mentioned in the introductory text of this response letter, new runs have been performed following both reviewers' comments. In specific, new runs to be used in the revised manuscript include May to September fire danger forecasts initialized in April (1-month lead) and March (2-month lead) that is one month and two months prior to the fire season, respectively, without spin-up and with spin-up performed from both SEAS5 and ERA5-Land data.

In the Methodology section of the revised manuscript, the new runs as well as a more analytical explanation of the spin-up together with a new figure presenting the different lead time and spin-up experiments will be provided as follows:

*It should be noted, that in order to commence the calculations of FWI, default initial values of FFMC, DMC, and DC are used. This means that a spin-up period is required to minimize the effects of errors in the initial conditions used in its calculation. Given that the longest time lag of the fuel moisture codes, as described above, is about 50 days, a spin-up period of up to two months is expected to be sufficient for both FWI and/or its subcomponents.*

*A fire season spanning from May to September (MJJAS), that coincide with the dry season in Attica according to the records of the Hellenic National Meteorological Service, is considered and six different experimental setups for FWI calculations are performed. In particular, SEAS5 MJJAS fire danger forecasts initialized in March and April (two months and one month in advance of the target fire season, respectively) without as well as with spin-up using SEAS5 and ERA5-Land data have been performed. In the case of spin-up, in 1-month (2-month) lead time forecasts, the FWI time series for April (March) have been calculated for the index to stabilize and were then removed from the analysis.*

[Figure]

Figure R3: Experimental setups used for FWI calculations in the current study. Forecasts are initialized in April (1-month lead time, in yellow) and March (2-month lead time, in red) while three different experiments concerning the spin-up period (a) with no spin-up (dashed line), (b) with spin-up implanting the ERA5-Land (solid line-circle symbol) and (c) with spin-up using the SEAS5 model data (solid line-diamond symbol) are shown.

**11. Lines 117 – 123: When explaining forecast lead-times and methods a figure with a scheme is usually very useful for the reader to understand the methodology. Can you include one?**

**Answer:** Please see the answer in comment 10 above.

**12. Line 135: "while in a second step of bias correction is applied". Please improve language.**

**Answer:** In the revised manuscript this sentence will be rephrased as follows:

*"In particular, the seasonal forecast meteorological variables used to calculate FWI are initially regridded to the ERA5-Land grid by means of bilinear interpolation and following, bias correction is applied using the empirical quantile mapping (EQM)."*

**13. Figure 1: More adequate as supplementary material.**

**Answer:** Figure 1 will be moved to Supplementary material following reviewer's comment.

**14. Figures 8 – 9: The meaning of the size of circles should be written in the caption.**

**Answer:** The following explanation will be added in the captions of the reliability diagrams in the revised version of the manuscript: *"The size of the points represents the number of forecasts falling in each bin."*

**15. Figure 10: "Annual number of fires (NOF) in Attica per year…". Please, remove "per year".**

**Answer:** In the updated figure of the revised manuscript, "per year" will be removed.

**16. Conclusions: Too much repetition of results.**

**Answer:** Based on the reviewer's comment, this section will be rewritten focusing more on the main findings of the analysis, conclusions and future work.

---

## Author Response (AR2)

We highly appreciate the reviewers' insightful and helpful comments, and we have included most of their suggestions in the revised manuscript. Based on both reviewers' comments new runs have been performed, therefore, the methodology as well as the results section were revised accordingly. Moreover, a separate discussion section was included together with a supplement with some of the plots/maps as suggested.

The new runs performed, include May to September fire danger forecasts initialized in April (1-month lead time) and March (2-month lead time) with no spin-up and with spin-up performed from both SEAS5 and ERA5-Land data. Finally, it should be noted that in the revised version the computations were performed and the respective maps/plots were constructed only for the Attica region, in order to minimize the computational cost.

**Referee #1**

**This work aims at providing high resolution probabilistic seasonal FWI forecasts for Attica (Greece) and verifying these forecasts using probabilistic verification measures for skill assessment (ROC skill score and reliability diagrams). To accomplish that, the authors compute FWI and its components for the fire season MJJAS using the dynamic forecasting system SEAS5 with lead times of 0 and 1 month (issued in May and April, respectively). The manuscript describes innovative research with potential operationality in a country that has been recently affected by large wildfires. The main issue with the manuscript is the almost non-existent discussion. Indeed, the majority of the problems found in the manuscript can be solved with the disentanglement of the section "Results and Discussion" into two sections. Furthermore, the paper has many typos and sentences that are not well written. I urge the authors to carefully re-read everything. Thus, since some of the revisions may take some time to tackle, I suggest major revisions.**

**Comments:**

1. **Line 96: You say that "FWI represents the frontal fire intensity…". I do not agree with this sentence. Frontal fire intensity is the energy output rate per unit length of fire front. FWI is not that. Can you please elaborate on what you intended to say? Also, I think the explanation of the FWI components is very loose, especially because you introduce FFMC, DMC, and DC explaining what they are but never defining what the abbreviatures mean. Indeed, lines 88 – 91 are not reader-friendly. Moreover, in line 92 when you introduce ISI and BUI for the first time you should put in parenthesis "(Initial Spread Index)" and "(Build-up Index)". You only do this on the second appearance.**

**Reply:**

**You say that "FWI represents the frontal fire intensity…". I do not agree with this sentence. Frontal fire intensity is the energy output rate per unit length of fire front. FWI is not that. Can you please elaborate on what you intended to say?**

The definition used in the manuscript is based on van Wagner (1987) who stated that FWI represents the intensity of the spreading fire as energy output rate per unit length of fire front (please refer to pp. 3 of the above-mentioned publication). Moreover, according to Wotton (2009), FWI is analogous to Byram's fireline intensity (Byram 1959) and it provides a general summary of fire weather and fuel moisture in a region when a single indicator of general fire potential is needed, for instance when communicating fire danger to the general public.

Therefore, in the revised manuscript (lines 114-115 of the marked-up manuscript) the FWI definition was complemented with the above-mentioned references as follows:

*"FWI represents frontal fire intensity (van Wagner, 1987) and can be used as a general index of fire danger (Wotton, 2009)."*

**Also, I think the explanation of the FWI components is very loose, especially because you introduce FFMC, DMC, and DC explaining what they are but never defining what the abbreviatures mean. Indeed, lines 88 – 91 are not reader-friendly. Moreover, in line 92 when you introduce ISI and BUI for the first time you should put in parenthesis "(Initial Spread Index)" and "(Build-up Index)". You only do this on the second appearance.**

A tighter as well as analytical description (mainly for the three moisture sub-indices) of each FWI subcomponent is provided in the revised manuscript (lines 94-107 of the marked-up manuscript) together with the explanation of the different abbreviations as follows:

*"The FWI system consists of six components each measuring a different aspect of fire danger (van Wagner, 1987). The first three primary sub-indices are fuel moisture codes, which are numeric ratings of the moisture content of the forest floor and other dead organic matter. The Fine Fuel Moisture Code (FFMC) is a numeric rating of the moisture content of litter and other cured fine fuels. FFMC is an indicator of the relative ease of ignition and the flammability of fine fuel, having a fast response to weather variations (approximately 0.5 days under "standard" conditions, i.e., noon temperature 25°C, relative humidity 30% and wind speed 10km/h). The Duff Moisture Code (DMC) is a numeric rating of the average moisture content of loosely compacted organic layers of moderate depth. This code gives an indication of fuel consumption and is characterised by a medium-term response to weather variations (approximately 10 days). The Drought Code (DC) is a numeric rating of the average moisture content of deep, compact organic layers. DC has a long-term response (about 50 days) to weather variations and is a useful indicator of seasonal drought effects on forest fuels, as well as the amount of smoldering in deep duff layers and large logs.*

*The two intermediate sub-indices, Initial Spread Index (ISI) and Build-Up Index (BUI), are fire behaviour indices. The ISI is a numerical rating of the expected fire rate of spread which combine the effect of wind and FFMC. The BUI is a numerical rating of the total amount of fuel available for combustion that combines the DMC and the DC."*

2. **Line 113: Indeed, FWI is to be computed at local time noon. I searched for Greek time, and in summer there is a difference of three hours to 12UTC, i.e., you are computing FWI at 15 local time. As I understand SEAS5 sub-daily values are available in a 6-hour window, which means you had to choose between using values at 9 or 15 local time. It is of paramount importance that you explain this here. Also, your discussion section must take this issue in consideration.**

**Reply:**

FWI is computed using meteorological records at ''noon local standard time" (Stocks et al, 1989). However, when using global long-range forecasts and climate models the appropriate temporal resolution for the input variables to the FWI is not always available therefore, proxies of noon local time are usually used. The choice of 12UTC as a proxy for noon conditions in the Mediterranean and Greece has been proposed in several studies before (e.g., Bedia et al., 2012, 2014, 2018; Herrera et al., 2013, Papagiannaki et al. 2020), therefore it seems a reasonable choice for our study too.

Moreover, according to van Wagner et al. (1987), FWI represents fire danger potential at its midafternoon peak (generally specified as 1600 hours). This is due to the fact that the noon variables in the original FWI System were correlated with fine fuel moisture and test fire data taken later in the afternoon. Finally, according to Papagiannaki et al. (2020), the meteorological conditions at 15 LST are highly conductive for the occurrence and spread of fires and the respective fire danger predictions are considered to be particular useful from an operational perspective based on the experience of the officers of the Hellenic Fire Service. Taking all the above into consideration, 15 LST instead of 9 LST input meteorological data were considered more appropriate to calculate and depict the noon /midafternoon peak of the index and fire activity in the area of interest. The following text was included in the methodology part of revised manuscript (lines 132-137 of the revised marked-up manuscript):

*"The 12 UTC was used as a proxy for local noon values required as input to FWI as proposed by several previous studies for the Mediterranean and Greece (e.g., Bedia et al., 2012, 2018; Herrera et al., 2013; Papagiannaki et al., 2020). Additionally, according to Papagiannaki et al. (2020), during the fire season the meteorological conditions at 12 UTC (i.e., 15 LST) are highly conductive to the occurrence and spread of fires as corroborated by the Hellenic Fire Service, thus, the respective fire danger predictions are considered to be particularly useful from an operational perspective."*

3. **Results and Discussion: Sections 3.1 and 3.2 are mainly a description of the results with a somewhat poor discussion. I recommend the authors to disentangle this section into two separate sections (3 Results; keeping 3.1 and 3.2; and 4 Discussion; 5 Conclusions). Moreover, the authors show a total of 9 figures in the results (and 1 in the methods), which seems too much for a 2-page section (results and discussion). I believe that with a stronger discussion section the 9 figures can better fit the amount of text. However, I recommend the authors to verify if all the figures are essential to the main body of the**

**manuscript (if not, send some of them to supplementary material). For example, Figure 1 is not essential. Above I recommended the authors to design a scheme, which may also be present in supplementary material (for readers who are new to the field understand the methodology of forecasts and lead-times).**

Reply: As suggested by the reviewer, in the revised manuscript two separate sections for Results and Discussion are provided. In the *Results* section the structure remains the same as in the first manuscript (keeping 3.1 and 3.2), while in the *Discussion* section the following are discussed: a) the prediction skill of the single meteorological variables used in the FWI system, b) the impact of lead-time and spin-up on fire danger forecast performance and c) the qualitative evaluation of fire danger forecasts to predict fire occurrence based on fire statistics. Part of the discussion is provided in Answer 5 below. Moreover, a supplement is provided with some of the figures that are complementary in our analysis (such as Fig. 1, Figs. 8-9 of the first manuscript). Finally, a new figure presenting the different lead time and spin-up experiments has been added in the methodology section. Please see also the answer to comment 10 below.

4.  **Lines 280 – 285: It is crucial to understand that FWI is a fire risk index, which only gives a picture of how susceptible a region is to burn. Without an ignition and vegetation prone to burn there will be no fire. It is interesting to see that it is possible to relate FWI forecasts for MJJAS fire season issued in April and May with the number of fires, but I do not think this is a key issue. However, doing such an analysis, why didn't the authors choose Fire Radiative Power (FRP), which could give an aggregated vision of the fire intensity in the region for the period of study? Wouldn't it be a more interesting variable than the number of fires?**

**Reply:**

As mentioned in Wotton (2009) FWI can be used as a general fire danger indicator. As it relies only on weather forcings, the index describes dangerous weather conditions conductive to uncontrollable fires in a region and not the susceptibility of a given region to forest fires which is highly dependent on other factors such as fuel load, fuel type and connectivity, topography etc. Thus, high fire danger can still be recorded where fires are otherwise inhibited due to e.g., fuel availability or the absence of an ignition.

Despite this limitation, FWI and its subcomponents have already been shown in several studies, to correlate well with fire activity data. In the recent study of Galizia et al. (2021) for European fire regimes, statistically significant moderate/strong correlations were found between FWI/ISI and the number of fires for fire-prone pyro-regions in Europe, with Greece (and Attica) categorized as such. Focusing on Greece, Dimitrakopoulos et al. (2011) found strong correlations between FWI and fire occurrence data, while Karali et al. (2014) found strong relationships between the number of fires and region-specific fire occurrence FWI thresholds.

Moreover, we are aware that recent studies (e.g., Durao et al., 2022) have used FRP/FRE values in order to assess fire intensity. However, our goal in this part of the analysis, is to provide an

evaluation of the ability of FWI hindcasts to predict actual fire occurrence and not fire intensity in Attica region, given the strong correlations between fire events and FWI (and its subcomponents) found in previous studies and described above.

5. **Figures 3 – 4: More discussion on these figures is needed. The authors need to develop on why fire season wind speed is better forecasted in March than in April. Why does precipitation show no skill? Also, it is very hard to relate the colours to values. I recommend you use one colour in steps of 0.2 and try to use more contrasting tones of reds and blues (this applies to Figures 2 – 6).**

**Reply:**

In Figure R1 below, the new maps depicting the ROCSS for the upper tercile category of air temperature and wind speed, as well as the lower tercile category of relative humidity and total precipitation for the 1-month and 2-month lead times are presented. After the reviewer's comments the colorbar/colorstep has been changed.

From Fig. R1 is evident that forecasts of relative humidity and wind speed (mainly for 1-month lead time) exhibit high discrimination skills, temperature exhibits low skill almost for the entire domain of study, while precipitation showed no skill for both lead time experiments. It is also evident that the forecast performance declines as the forecast lead time (i.e., the period between the target fire season and the initialization date of the forecast) increases. This is in line with previous studies by Doblas-Reyes et al. (2013) and van den Hurk et al. (2012).

According to our literature search, the Mediterranean region is an area of transition between subtropical and mid-latitudes, where seasonal forecasts are challenging, therefore the assessment of the added value and the limitations of these products is of paramount importance when developing climate services (Calì Quaglia et al., 2022). The same study found statistically significant temperature anomaly correlations over the eastern Mediterranean between the SEAS5 and the ERA5 reference dataset, however, summer ROC skill score was not discussed in that study. Additionally, summer precipitation showed limited skill, located mainly at the western part of the Mediterranean. In general, the climate of the western Mediterranean is more predictable than the eastern part of the domain, probably due to the influence of El Niño-Southern Oscillation (ENSO) and North Atlantic Oscillation (NAO) teleconnections (Calì Quaglia et al., 2022; Frías et al., 2010). Concerning relative humidity, our results are in line with previous studies (Bedia et al., 2018; Bett et al., 2018) who found significant skills over the eastern Mediterranean using the ECMWF System 4 forecasting system. Finally, wind speed can be considered a promising variable regarding skill, as it is more closely related to the larger-scale atmospheric circulation than more complex processes like precipitation. According to Bett et al. (2022), the wind skill was found to be patchy throughout Europe especially during summer using however the Sys4 forecasting system. Focusing on Attica where our results present high skill regarding the wind, Kassomenos (2010) found the Etesians (dry north winds prevailing during summer) to be very often associated with the development of extreme wildfires in Greece, while Pashalidou et al. (2016), pointed out that

mesoscale and local systems can also play an important role on fire development, as they interact with, and may exacerbate the larger scale circulation patterns.

The above was included in the discussion part of the revised manuscript (Section 4.1).

[Figure]

*Figure R1: ROC Skill Scores of the FWI input variables for 1-month and 2-month lead time MJJAS forecasts that correspond to high fire danger values, i.e., upper tercile of air temperature (T2M), upper tercile of wind speed (WSS), lower tercile of air relative humidity (RH) and lower tercile of total precipitation (PR). The grid points with significant ROCSS values are indicated by circles (α=0.05).*

**6. Line 27: "as regards" to "regarding".**

**Reply:** "As regards" was replaced with "regarding" to the revised manuscript following the reviewer's comment.

**7. Line 84: When defining FWI you should add the Van Wagner reference.**

**Reply:** According to the reviewer's suggestion the reference to van Wagner (1987) was added in FWI definition (please see also the answer to the 1st comment).

**8. Line 116: replace "temperature" by "temperatures".**

**Reply:** "Temperature" was replaced with "temperatures" in the revised manuscript.

**9. Line 117: Why May – September as fire season? Why do you include May and ignore October? Is this common for Greece?**

**Reply:** It is true that according to the Hellenic Fire Service the official fire season for the whole Greece is the period from May to October. However, in the current study which focuses on the Attica region, in order to define the fire season, we selected the region's particular dry season, which, according to the climatological records of the Hellenic National Meteorological Service (HNMS), covers the months from May to September as can be seen in the following figure (Fig. R2).

[Figure]

*Figure R2: Monthly mean precipitation (red line) and number of days with precipitation /wet days (blue columns) for the period 1955-2010 for three indicative stations in Attica region.*

**10. Line 118: Why do you stop at 1-month lead time? Wouldn't it be interesting to see the predictive skill of forecasts starting in March? I believe 6-month lead times (7 months prior to target) are available, which means that for September there are forecasts initialized in March. Does it have to do with the spin-up period? Moreover, please try to better explain what a spin-up period is.**

**Reply:**

As has already been mentioned, new runs have been performed following both reviewers' comments. In specific, new runs include May to September fire danger forecasts initialized in April (1-month lead) and March (2-month lead) that is one month and two months prior to the fire season, respectively, without spin-up and with spin-up performed from both SEAS5 and ERA5-Land data.

In the Methodology section of the revised manuscript, the new runs as well as a more analytical explanation of the spin-up together with a new figure (Fig. R3) presenting the different lead time and spin-up experiments are provided as follows (lines 141-151 of the marked-up manuscript):

*"It should be noted, that in order to commence the calculations of FWI, default initial values of FFMC, DMC, and DC are used. This means that a spin-up period is required to minimize the effects of errors in the initial conditions used in its calculation. Given that the longest time lag of the fuel moisture codes, as described above, is about 50 days, a spin-up period of up to two months was considered sufficient for both FWI and/or its subcomponents.*

*A fire season spanning from May to September (MJJAS), that coincides with the dry season in Attica according to the records of the Hellenic National Meteorological Service, was considered and six different experimental setups for FWI calculations were implemented. In particular, we performed SEAS5 MJJAS fire danger forecasts initialized in March and April (two months and one month in advance of the target fire season, respectively), without and with spin-up, using both SEAS5 and ERA5-Land data. In the case of spin-up, in 1-month (2-month) lead time forecasts, the FWI time series for April (March) were firstly calculated for the index to stabilize and were then removed from the analysis."*

[Figure]

*Figure R3: Experimental setups used for FWI calculations in the current study. Forecasts are initialized in April (1-month lead time, in yellow) and March (2-month lead time, in red) while three different experiments concerning the spin-up period (a) with no spin-up (dashed line), (b) with spin-up implanting the ERA5-Land (solid line-circle symbol) and (c) with spin-up using the SEAS5 model data (solid line-diamond symbol) are shown.*

**11. Lines 117 – 123: When explaining forecast lead-times and methods a figure with a scheme is usually very useful for the reader to understand the methodology. Can you include one?**

**Reply:** Please see the answer in comment 10 above.

**12. Line 135: "while in a second step of bias correction is applied". Please improve language.**

**Reply:** In the revised manuscript this sentence was rephrased as follows (lines 169-171 of the marked-up manuscript):

*"In particular, the seasonal forecast meteorological variables used to calculate FWI were initially regridded to the ERA5-Land grid by means of bilinear interpolation and next, bias correction was applied using the empirical quantile mapping (EQM)."*

**13. Figure 1: More adequate as supplementary material.**

**Reply:** Figure 1 was moved to Supplement following reviewer's comment.

**14. Figures 8 – 9: The meaning of the size of circles should be written in the caption.**

Reply: The following explanation was added in the captions of the reliability diagrams in the revised version of the manuscript (in Supplement): *"The size of the points represents the number of forecasts falling in each bin."*

**15. Figure 10: "Annual number of fires (NOF) in Attica per year…". Please, remove "per year".**

Reply: In the updated figure of the revised manuscript (Fig. 7), "per year" was removed.

**16. Conclusions: Too much repetition of results.**

Reply: Based on the reviewer's comment, this section was shortened focusing more on the scope, conclusions and future work.

**Referee #2**

**This work examines the utility of probabilistic seasonal forecasts from the fifth generation ECMWF system combined with the Canadian FWI index for fire season forecasts over Greece, with a focus on the Attica region. The results are potentially of high value, given that this region is prone to regular fires. The general approach makes sense and the results are analysed using good quality standard assessment methods which give consistent results.**

**I have two main points, which relate to potentially improving forecast skill, rather than the quality of the study per se:**

**1) I'm not sure how the Greek fire service plans resource allocation, but, rather than attempting an aggregate forecast for the entire fire season, would it not be useful to, say, divide the fire season in two, and give forecasts for each half separately (e.g. for may-july initialised in march/april; and for july-sep initialised in may/june). This would allow forecasts with shorter lead times, which should in turn improve skill.**

**2) Related to this: the question of why the forecast skill seems to be so low for the longer-timescale components of the FWI system (those for the denser fuels). I guess this arises from two things: if I understand correctly, the authors do not use observations to spinup the FWI system. Since the BUI and DC have spinup timescales of the order of 15 and 50 days, so**

**initialisation with obs would surely give some additional predictability for the latter in particular. This would be more relevant if my suggestion 1 is implemented.**

**Reply:**

Comments 1 and 2 are somewhat related, thus, the experiments described below have been performed in order to examine whether the division of the May-June-July-August-September (MJJAS) period in two sub-periods, May-June-July (MJJ) and July-August-September (JAS) improves the predictability of FWI and/or its subcomponents:

• SEAS5 forecasts initialized in March and April for MJJ (two month and one month in advance of the target fire season, respectively) with no spin up and spin up performed from both SEAS5 and ERA5-Land data

• SEAS5 forecasts initialized in May and June for JAS (two month and one month in advance of the target fire season, respectively) with no spin up and spin up performed from both SEAS5 and ERA5-Land data

In all experiments bias correction was applied using daily data for the period of interest (MJJ, JAS) using a moving window width of 31-days, following the methodology of the first version of the manuscript. In addition, the results of the statistically downscaled temperature, relative humidity, wind speed and precipitation for each period are also presented. It should be noted that in the revised version the computations were performed, and the respective maps/plots were constructed only for the Attica region in order to minimize the computational cost.

Regarding MJJ, the ROCSS for the upper tercile category of air temperature and wind speed, as well as the lower tercile category of relative humidity and total precipitation for the 1-month and 2-month lead times are presented in Fig. R1. From Fig. R1 it is evident that forecasts initialized in April (1-month lead time) exhibit higher discrimination skill for temperature and wind speed, while forecasts initialized in March (2-month lead time) exhibit higher discrimination skill for relative humidity and precipitation. As far as the FWI is concerned, from the tercile plots (Fig. R2) is evident that 2-month lead time forecasts exhibit higher skill than the corresponding 1-month lead time. The highest upper tercile ROCSS, 0.7 is found for the 2-month lead time forecasts and when using ERA5-Land for the spin-up, classified to perfect reliability (according to the new tercile plots-not shown) while the rest of the experiments are classified as marginally useful (not shown).

Longer spin-up periods and implanting ERA5-Land in the spin-up, increases the DC upper tercile ROCSS compared to 1-month lead time spin-up (Fig. R3) with the predictions classified as marginally useful (not shown). In contrast, the highest BUI upper tercile ROCSS is found for the 2-month lead time forecast with spin-up from SEAS5 (Fig. R4) with the predictions classified as marginally useful (not shown). Concerning DMC, the highest upper tercile ROCSS is found for the 2-month lead time forecasts without spin-up (0.58) and with spin-up (0.49) when using SEAS5 data, while the predictions are classified as marginally useful (not shown). Finally, ISI results indicate that the specific sub-component is insensitive to the spin-up and its results are mostly controlled by the ROCSS of the meteorological variables used for the FWI calculations from the

different lead time forecasts. In particular, from Fig. R6 it is evident that the highest ISI upper tercile ROCSS, 0.83 (perfect reliability), is found for all 1-month lead time experiments (with and without spin up) while lower ROCSS, 0.66 (marginally useful), is found for the corresponding 2-month lead time experiments. The highest ROCSSs are attributed to the higher discrimination power of temperature and wind speed over the area under study in the 1-month lead time forecasts.

For the JAS period the results are not found as encouraging as found in MJJ. In particular, from Figure R7 it is evident that both forecasts initialized in June (1-month lead time) and May (2-month lead time) exhibit very low discrimination skill for the majority of the meteorological variables used to drive the FWI calculations with the exception of high discrimination skill shown for relative humidity for 2-month lead forecasts and the relatively low skill shown for wind speed for both forecasts. As a result, poorer skill is found for FWI and ISI in JAS (Figs. R8 and R9, respectively) compared to MJJ, with the predictions for the 1-month lead time forecast characterized as dangerously useless and as not useful for FWI and ISI, respectively (not shown), while for the 2-month lead time forecast the predictions are characterized as dangerously useless and not useful (not shown) for the same components, respectively.

From the above analysis, we conclude that both the initialisation date of the forecasts, the length of the spin-up period as well as the way spin-up is implanted (with or without ERA5-Land) play an important role in the prediction of FWI and its sub-components. Therefore, in the revised manuscript we maintained the MJJAS period but analysed and discussed the experiments as above (i.e., 1-month and 2-month lead time fire danger experiments with no spin-up and spin-up performed from both SEAS5 and ERA5-Land data).

[Figure]

*Figure R1: ROC Skill Scores of the FWI input variables for 1-month and 2-moth lead time MJJ forecasts that correspond to high fire danger values, i.e., upper tercile of air temperature (T2M), upper tercile of wind speed (WSS), lower tercile of air relative humidity (RH) and lower tercile of total precipitation (PR). The grid points with significant ROCSS values are indicated by circles (α=0.05).*

[revised manuscript text omitted]

**-----------minor points**

**The reliability diagrams are useful in that they're an alternative way of valdiating the forecasts, but perhaps could be in supplementary material, as they seem to largely just backup the ROCSS results.**

**Reply:** Following the reviewer's suggestions all reliability diagrams were moved to Supplement.

**I find the LM0/LM1 acronyms rather unnecessary and confusing. Suggest using e.g. '1 month lead' as it's not much longer, and much clearer.**

**Reply:** Acronyms were changed following the reviewer's suggestion.